# Study on the Correlation between Magnetic Field Structure and Cold Electron Transport in Negative Hydrogen Ion Sources

**Mengjun Xie** **, Dagang Liu \*, Huihui Wang and Laqun Liu**

School of Electronic Science and Engineering, University of Electronic Science and Technology of China, Chengdu 610054, China; 201711040128@std.uestc.edu.cn (M.X.); huihuiwang@uestc.edu.cn (H.W.); liulq@uestc.edu.cn (L.L.)
\* Correspondence: ldg12345@uestc.edu.cn

**Abstract:** In most negative hydrogen ion sources, an external magnet is installed near the extraction region to reduce the electron temperature. In this paper, the self-developed CHIPIC code is used to simulate the mechanism of a magnetic filter system, in the expansion region of the negative hydrogen ion source, on "hot" electrons. The reflection and the filtering processes of "hot" electrons are analyzed in depth and the energy distribution of electrons on the extraction surface is calculated. Moreover, the effects of different collision types on the density distribution of "cold" electrons along the X-axis and the spatial distribution of "cold" electrons on the X−Z plane are discussed. The numerical results show that the electron reflection is caused by the magnetic mirror effect. The filtering of "hot" electrons is due to the fact that the magnetic field constrains most of the electrons from reaching the vicinity of the extraction surface, being that collisions cause a decay in electron energy. Excitation collision is the main decay mechanism for electron energy in the chamber. The numerical results help to explain the formation process of "cold" electrons at the extraction surface, thus providing a reference for reducing the loss probability of $H^-$.

**Keywords:** negative hydrogen ion source; particle in cell; Monte Carlo collision; magnetic mirror effect

## 1. Introduction

Compared with other multi-type ion sources, the radio frequency (RF) negative hydrogen ion source has a relatively simple structure, reliable performance, long life, and is almost maintenance-free [1]. It has its unique advantages and characteristics. Therefore, the RF negative hydrogen ion source is favored by many researchers. In 2007, the RF negative hydrogen ion source was established as the reference scheme for plasma generation in the International Thermonuclear Experimental Reactor (ITER) neutral beam injection system [2–5]. With the development and testing of the ITER neutral beam test facility located in Padova, Italy, countries are experiencing a research boom with respect to negative hydrogen ion sources [6–9]. At present, the Max-Planck-Institute für Plasma Physik (IPP) in Germany is one of the leading institutions in the research of RF negative hydrogen ion sources [10,11]. Some experimental parameters obtained by the BATMAN, BATMAN upgrade, MANITU, ELISE, and RADI experimental devices [12–14] have reached or exceeded the requirements of the ITER project. Based on the BATMAN testbed, India has developed an ion source device, ROBIN [15], similar to the BATMAN structure. China has also successively developed a single-exciter experimental device, HUST [16], a RF negative ion source device, HUNTER [17], and so on. At the same time, a large number of numerical simulation studies have also been carried out on negative hydrogen ion source devices, mainly using the two-dimensional or three-dimensional Particle-In-Cell/Monte Carlo Collision (PIC/MCC) method to perform a lot of work on $H^-$ volume generation, surface generation processes, and extraction processes [18–20]. The factors affecting the

beam quality have been studied in an attempt to clarify the physical mechanism in the source and to improve the performance of the device.

Many examples in the existing literature show that the negative hydrogen ion source needs to add a magnetic filter field near the extraction region to filter the "hot" electrons ($T_e \geq 10$ eV), so as to reduce the electron temperature and reduce the stripping loss of negative hydrogen ions [21–24]. The design of the magnetic filter field is strongly dependent on experiments [23]. In addition to the effect of "reducing the temperature of electrons", which is widely understood by researchers, the mechanism behind it also needs to be studied by means of numerical calculation. Based on the three-dimensional PIC/MCC method [25] and using the CHIPIC code [26–28], this paper studies the transport mechanism of "hot" electrons passing through the magnetic filter field system in the expansion region and the influence of different collision types on "hot" electron filtering. Results from modelling [29,30] show that a gradient drift occurs in the IPP prototype source. Likewise, the vertical drift of electrons due to gradient drift is observed in this paper. The temperature of the initial electrons is reduced from 10 eV to about 3.2 eV through the magnetic filter field system, which is consistent with the 3 eV of a reference [31] with the same magnetic field structure. However, numerical results show that the magnetic filter field filters both "hot" and "cold" electrons (energy $\varepsilon \leq 2$ eV), which is different from the descriptions in the literature. The filtering, reflection, and drift phenomena of electrons in the numerical results are presented and discussed to understand the complex behavior of electrons on the magnetic filter field. The research in this paper helps elucidate the particle transport process under the magnetic filter field system, the formation process of cold electrons on the extraction surface, and the objective role of the magnetic filter field system in negative hydrogen ion source devices.

## 2. Simulation Model and Methods

Figure 1 is a schematic diagram of a typical negative hydrogen ion source structure. It is mainly divided into three parts: the driver, the expansion region, and the extraction region. The driver generates hydrogen or deuterium plasma through RF power coupling (up to 100 kW at 1 MHz) with an electron temperature greater than 10 eV and an electron density greater than $10^{18}$ m$^{-3}$. As the plasma diffuses into the expansion region, the electron temperature is cooled below 2 eV by the magnetic filter field [21]. The extraction region mainly includes a bias plate (BP) and three grids (a plasma grid PG, an extraction grid EG, and a grounded grid GG). There is usually a positive bias voltage of about 15 V between BP and PG. The purpose of this paper is to study the mechanism of the magnetic filter field on "hot" electrons, so the simulation domain selects the expansion region, which contains a long-range weak magnetic filter field. The surface production process and the extraction and acceleration processes of $H^-$ ions are not considered, so the model we built only simulates the expansion region of the front end of BP shown in Figure 2. Since BP is connected to the source body, the expansion region can be considered to be equipotential. The main generation method for $H^-$ ions in the expansion region is volume production, that is, $H_2$ achieves an excited state after colliding with fast electrons ($T_e \geq 5$ eV) and then dissociates and adsorbs with low-temperature electrons to generate $H^-$.

Figure 2 is the schematic diagram of the X$-$Z cross section of the negative hydrogen ion source expansion region model. The simulation domain is 18 cm $\times$ 60 cm $\times$ 30 cm. The black frame represents the metal conductor; the yellow lines represent the ends of the emission surface of the electron beam and the BP, which are all metal conductor materials; the green dashed line represents the ends of the extraction surface; and the grey area represents the plasma. Samarium-cobalt permanent magnets are attached on both sides of the chamber. When electrons reach the simulation boundary (the surface of a metal conductor), they are destroyed. This paper approximates the process of generating plasma by an RF power coupling in the driver as follows: electron beams are emitted from a perfect conductor surface, colliding with the background gas to generate plasma, and the emitting surface is circular, with a radius of 12 cm.

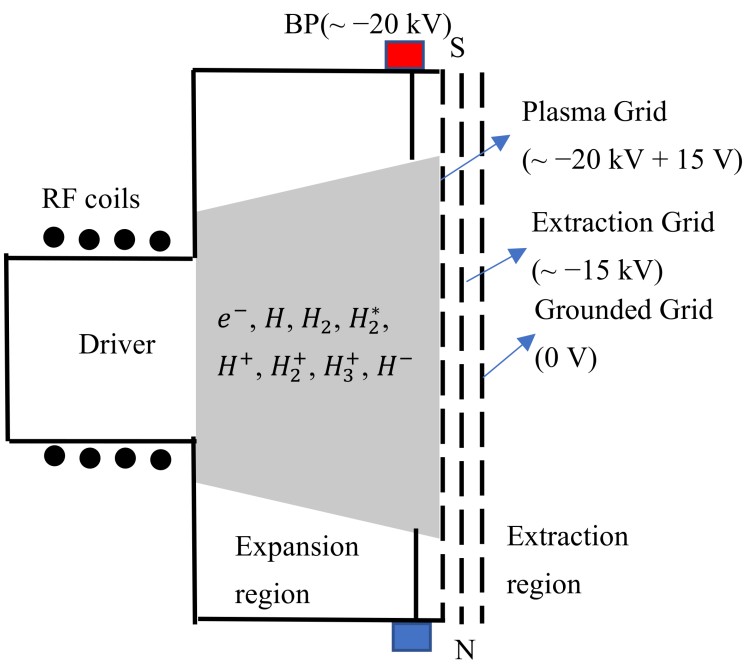

**Figure 1.** Schematic view of negative hydrogen ion source.

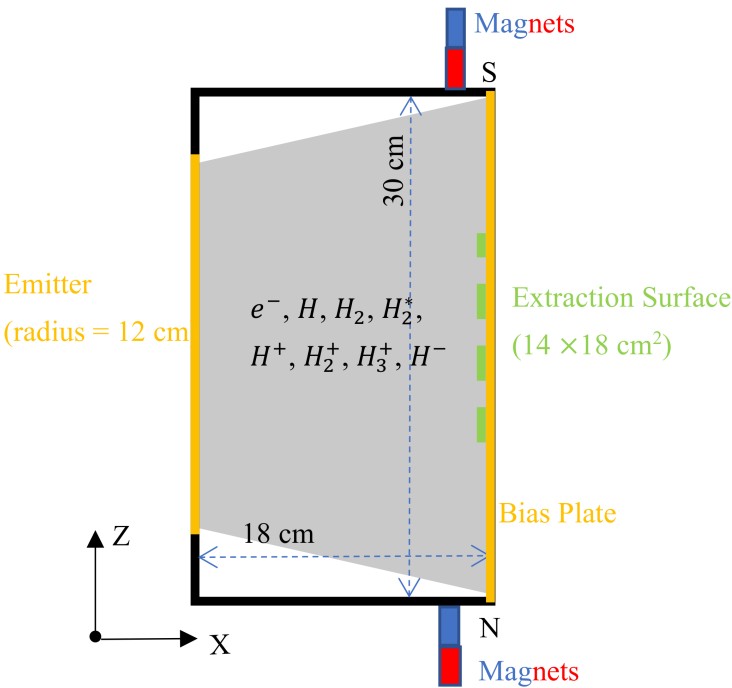

**Figure 2.** The schematic diagram of the X−Z cross section of the negative hydrogen ion source expansion region model.

The electron beam temperature is set at 10 eV, and the electron velocity distribution follows the Gaussian distribution. The initial electron density is $5 \times 10^{18}$ m$^{-3}$, and the background hydrogen has a pressure of 0.6 Pa and a temperature of 1200 K. Due to the extremely high initial density of electrons, affected by the space-charge-limited current, the generated reverse electric field force leads to most of the electrons being unable to be successfully emitted and thus accumulating on the emitting surface. Therefore, the model ignores the interactions between particles and adopts the electrostatic simulation method.

The simulation parameters are set as shown in Table 1 [32–34]. The main collision reactions considered in the expansion region are shown in Table 2 [32,35,36]. The magnetic filter field is calculated by the scalar magnetic potential finite-difference method (SMPM) [26,37]. Magnetic boxes are located on both sides of the source body, and there are 4 magnetic boxes in total. Each magnetic box could hold $2 \times 4$ samarium–cobalt magnets; that is, CoSm magnets are superimposed, and 2 CoSm magnets are placed on each layer for a total of 4 layers. The size of each magnet is 9 mm $\times$ 13 mm $\times$ 50 mm, the maximum magnetic field intensity is 1 T, and the magnetization direction is along the direction of "13 mm" [22].

**Table 1.** Simulation parameters used in code [32–34].

| Simulation Parameters | | |
|---|---|---|
| Hydrogen gas pressure | $P$ | 0.6 Pa |
| Hydrogen gas temperature | $T_g$ | 1200 K |
| Initial electron density on the emitter surface | $N_e$ | $5 \times 10^{18}$ m$^{-3}$ |
| Hydrogen gas density | $N_{H2}$ | $4 \times 10^{19}$ m$^{-3}$ |
| Atom to molecule density ratio | $N_H/N_{H2}$ | 0.2 |
| Density ratio of vibrationally excited to ground state hydrogen molecules | $N_{H2(v>3)}/N_{H2(v=0)}$ | 0.01 |
| Charged particle injection ratios | $e-: H+: H_2+: H_3+$ | 1:0.2:0.6:0.2 |
| Electron temperature | $T_e$ | 10 eV |
| Chamber wall potential | $\Phi$ | 0 V |
| Grid size | $\Delta x = \Delta y = \Delta z$ | 4 mm |
| Grid number | $N_x \times N_y \times N_z$ | 506,250 |
| Macro particle number of electron | $N$ | 2.2~2.7 $\times 10^6$ |
| CoSm magnet size | $x', y', z'$ | 9 mm $\times$ 50 mm $\times$ 13 mm |
| Extraction chamber size | $X, Y, Z$ | 18 cm $\times$ 60 cm $\times$ 30 cm |
| Timestep | $\Delta t$ | 0.1 ns |
| Simulation time | $T$ | 15 μs |

**Table 2.** Reaction in the transport simulation [32,35,36].

| Index | Collision Type | Collision Species | Threshold (eV) |
|---|---|---|---|
| 1 | Elastic | $e + H \rightarrow e + H$ | 0.01 |
| 2 | Ionization | $e + H^* \rightarrow 2e + H^+$ | 13.6 |
| 3 | Ionization | $e + H_2^* \rightarrow 2e + H_2^+$ | 15.42 |
| 4 | Dissociative Ionization | $e + H_2^* \rightarrow 2e + H + H^+$ | 21.1 |
| 5 | Dissociative Recombination | $e + H_3^+ \rightarrow 3H$ | 0.1 |
| 6 | Dissociative Recombination | $e + H_2^+ (0 \leq v \leq 9) \rightarrow H(1s) + H^*(n \geq 2)$ | 0.01 |
| 7 | Dissociative Attachment | $e + H_2^* (v > 3) \rightarrow H^- + H$ | 0.1 |
| 8 | Dissociation | $e + H_2\left(X^1\Gamma_g^+, v'' = 0\right) \rightarrow e + H(1s) + H^*(2s)$ | 14.9 |
| 9 | Vibrational Excitation | $e + H_2(v = 0) \rightarrow H_2(v = 1) + e$ | 0.895 |
| 10 | Vibrational Excitation | $e + H_2(v = 0) \rightarrow H_2(v = 2) + e$ | 1.38 |
| 11 | Electronic Excitation | $e + H_2\left(X^1\Gamma_g^+, v'' = 0\right) \rightarrow H_2^*\left(B^1\Gamma_u^+ 2p\sigma\right) + e$ | 12 |
| 12 | Electronic Excitation | $e + H_2\left(X^1\Gamma_g^+, v'' = 0\right) \rightarrow H_2^*\left(C^1\Pi_u 2p\pi\right) + e$ | 12 |
| 13 | Electronic Excitation | $e + H_2\left(X^1\Gamma_g^+, v'' = 0\right) \rightarrow H_2^*\left(b^3\Gamma_u^+\right) + e$ | 11.72 |
| 14 | Electronic Excitation | $e + H_2\left(X^1\Gamma_g^+, v'' = 0\right) \rightarrow H_2^*\left(a^3\Gamma_g^+\right) + e$ | 7.93 |
| 15 | Electronic Excitation | $e + H_2\left(X^1\Gamma_g^+, v'' = 0\right) \rightarrow H_2^*\left(c^3\Pi_u^+\right) + e$ | 11.72 |
| 16 | Electronic Excitation | $e + H_2\left(X^1\Gamma_g^+, v'' = 0\right) \rightarrow H_2^*\left(E, F^1\Gamma_g^+\right) + e$ | 15 |
| 17 | Electronic Excitation | $e + H_2\left(X^1\Gamma_g^+, v'' = 0\right) \rightarrow H_2^*\left(e^3\Gamma_u^+\right) + e$ | 17.5 |
| 18 | Dissociative Excitation | $e + H_2^+ (0 \leq v \leq 9) \rightarrow e + H^+ + H$ | 2.7 |
| 19 | Dissociation | $e + H_3^+ \rightarrow e + H^+ + 2H$ | 14 |

## 3. Numerical Results and Analysis

### 3.1. Magnetic Filter Field Simulation Result

In this paper, the scalar magnetic potential finite-difference method is used to calculate the magnetic filter field. The layout of the magnetic field is as referred to in [22,33]; that is, the magnets are located in the magnetic boxes on both sides of the chamber, forming a strong magnetic field on the two sides and weak one in the middle. The magnetic field line enters the expansion region from the chamber wall and exits from the other chamber wall, and it does not form a closed magnetic field line, which is a typical structure and characteristic of the negative hydrogen ion source magnetic filter field. Figure 3a shows the distribution of the magnetic filter field $B_z$ on the Y−Z plane calculated by SMPM. Figure 3b shows the calculation result for the distribution of the magnetic filter field $B_z$ along the X-axis. It shows that the maximum magnetic induction intensity $B_z$ along the X-axis at the center of the chamber is about 7.4 mT, which is in good agreement with the magnetic field intensity distribution curve in [33].

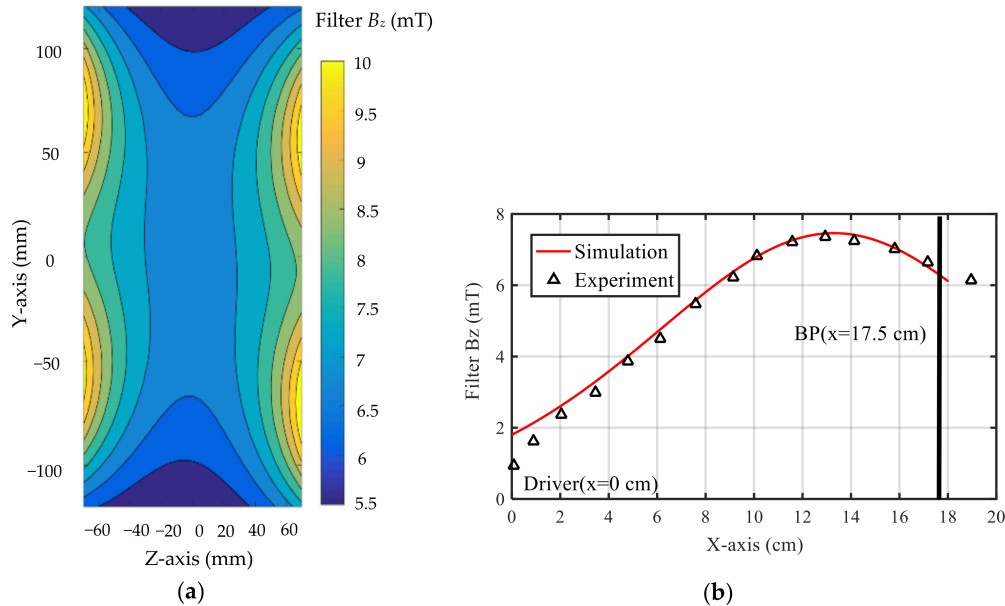

**Figure 3.** (**a**) The distribution of magnetic filter field $B_z$ calculated by SMPM on the Y−Z plane; (**b**) Magnetic filter field $B_z$ distribution along the X-axis compared with experimental data.

Section 3.1 is a collision-free model. The filtering effect of the magnetic field on the electron beam with a temperature of 10 eV on the X−Z plane is shown in Figure 4. It can be seen that most of the electrons are effectively filtered by the magnetic filter field, and only a small number of the electrons are allowed to pass through the expansion region to the extraction surface (as shown in Figure 2). The whole simulation time $T$ is 15 µs. The calculation is regarded as steady-state when the number of electrons in the chamber reaches saturation after 15 µs. When $T = 9 \sim 15$ µs, compared with the absence of a magnetic filter field, the proportion of extracted low-energy electrons ($\varepsilon \leq 2$ eV) increases from 2.6% to 10.3%, and the temperature of the extracted electrons decreases from 10 eV to 6.5 eV, but the number of extracted electrons drops from $1.95 \times 10^{17}$ to $5.37 \times 10^{13}$. This suggests that the magnetic filter field can effectively filter both "hot" and "cold" electrons. For the objects to be filtered, "hot" and "cold" electrons have equal statuses. Since collisions are not considered in this section, the electron energy does not decay in any form, and the mechanism of the magnetic filter field can be regarded as a matter of how electrons are transported in the magnetic filter field.

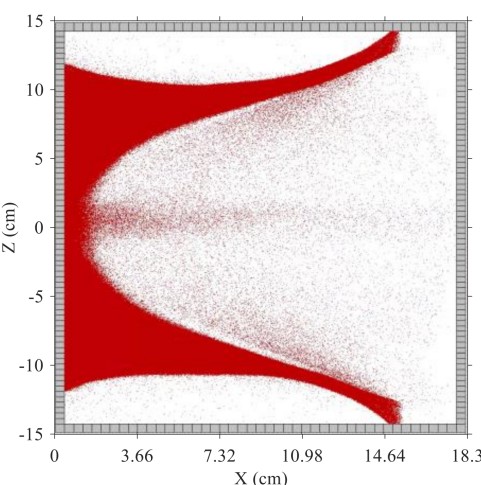

**Figure 4.** The filtering effect of the magnetic field on the electron beam with a temperature of 10 eV on the X−Z plane at 15 μs.

In order to explore the general law of electron transport paths in the expansion region, the electron emission position is fixed at the centre of each grid on the emission surface (i.e., the emitter in Figure 2) to observe the electron trajectory. The energy of all electrons is fixed at 30 eV. Results are shown in Figure 5. Figure 5a shows the projection of all electrons on the X−Z plane. Figure 5b is the projection of some electrons on the X−Z plane in the range of the Y-axis [−4 mm, 4 mm]. Electrons travel in the +X direction from the emission surface (at X = 0 cm). As the magnetic field intensity increases, the electrons gyrate along the magnetic field lines. Figure 5b clearly shows that some of the electrons are reflected by the strong magnetic field on both sides of the chamber as they pass through the expansion region. Such a phenomenon can be explained by the magnetic mirror effect; that is, having a strong magnetic field at both ends and a weak one in the middle causes charged particles to be reflected by the strong field, just as light is reflected by a flat mirror. The magnetic mirror phenomenon can be explained by Equation (1).

$$\frac{1}{2}mv^2 = \frac{1}{2}mv_\perp^2 + \frac{1}{2}mv_\parallel^2 = \text{constant} \tag{1}$$

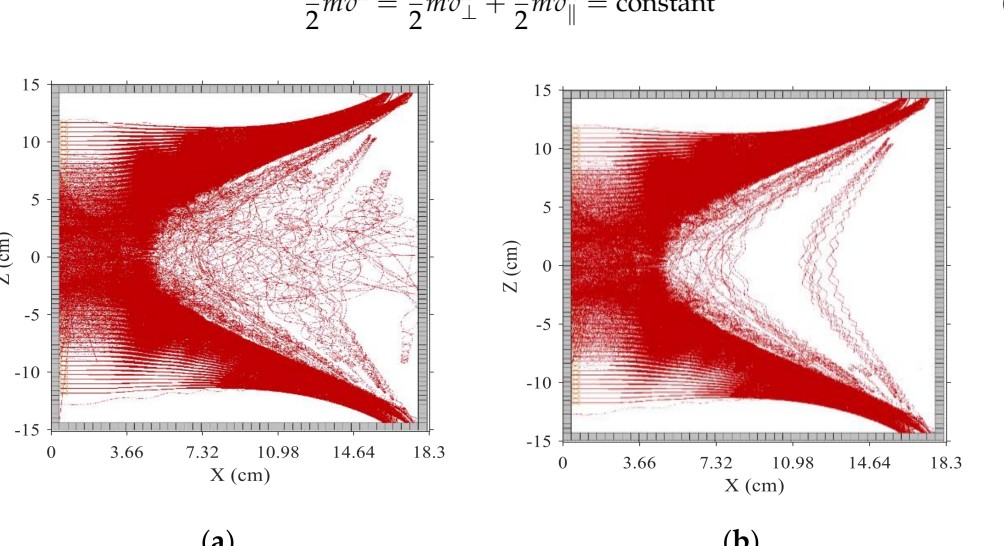

**(a)**  **(b)**

**Figure 5.** (**a**) The projection of all electrons on the X−Z plane; (**b**) The projection of electrons on the X−Z plane within the range of the Y-axis [−4 mm, 4 mm].

$v_\perp$ and $v_\parallel$ are the velocities of the electrons perpendicular to and parallel to the magnetic field, respectively, and $v$ is the velocity of the electrons in the direction of the resultant force. Since this model has no applied electric field, does not consider the interaction between electrons, and ignores the influence of gravity, the Lorentz force does not work, so the total energy of the electrons is conserved. When the magnetic field intensity $B$ increases, $v_\perp$ increases and $v_\parallel$ decreases. When $B$ is strong enough, $v_\parallel$ can decrease to zero (i.e., $v = v_\perp$) such that the longitudinal motion of some electrons in Figure 5b is suppressed, followed by reverse motion, which is continuously reflected in the chamber. Therefore, as shown in Figure 5, most of the electrons in the expansion region are still constrained by the magnetic filter field and move along the magnetic field lines to the chamber wall, while a few electrons are reflected by the strong magnetic field at both ends and move back and forth along the longitudinal direction of the chamber. With the effect of the magnetic filter field, the number of electrons guided to the chamber wall is greatly increased, and the number of electrons reaching the extraction surface is effectively reduced.

### 3.2. Calculation of Electron Energy Probability Function

In this section, the collision reactions shown in Table 2 and the simulation parameters in Table 1 are applied to the expansion region model. According to Equations (2) and (3) [38], the electron energy probability function (*EEPF*) diagram shown in Figure 6 is calculated.

$$EEDF(\varepsilon) = \frac{2}{kT}\sqrt{\frac{\varepsilon}{\pi kT}}\mathrm{e}^{-\frac{\varepsilon}{kT}}, \; EEPF(\varepsilon) = EEDF(\varepsilon)/\sqrt{\varepsilon} \tag{2}$$

$$lg\left[\frac{EEDF(\varepsilon)}{\sqrt{\varepsilon}}\right] = -\frac{1}{kTln10}\varepsilon + lg\left(\frac{2}{kT}\sqrt{\frac{1}{\pi kT}}\right) \tag{3}$$

where $\varepsilon$ is the electron energy (i.e., $\frac{1}{2}mv^2$). The distribution function of $\varepsilon$ is set to $EEDF(\varepsilon)$, and $KT$ is the electron temperature. As shown in Equation (3), the ratio of the absolute inverse of the slope of the *EEPF* straight line to $ln10$ is $KT$. $KT_{low}$ and $KT_{high}$ represent the electron temperatures of low and high energy states, which are 0.6 eV and 6.4 eV, respectively. It can be seen from Figure 6 that the electron energy shows a bi-Maxwellian electron energy distribution function (*EEDF* [39]), that is, a low temperature followed by a higher temperature, which is in line with the typical characteristics of negative hydrogen ion source discharge. The electron temperature of the extraction surface is calculated to be about 3.2 eV, which is consistent with the typical value of 3 eV as measured by a negative hydrogen ion source with the same magnetic field structure in [31]. In addition, the calculated low energy range at low temperature is 0–8 eV, which is consistent with the low energy range measured by Optical Emission Spectroscopy (OES) method and Langmuir Probe [31]. Therefore, the expansion region model is reasonably established, and the collision reactions and types shown in Table 2 basically conform to the main electron collision reactions of the negative hydrogen ion source.

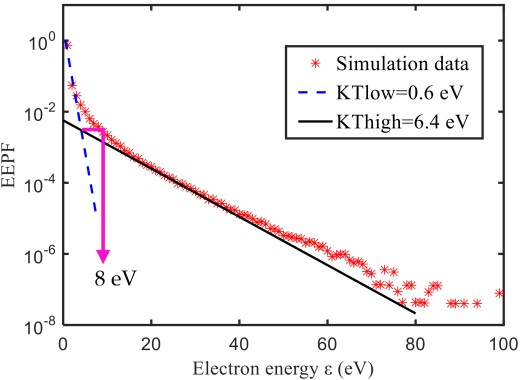

**Figure 6.** Diagram of EEPF as a function of electron energy.

### 3.3. Influence of Different Collision Reactions on Electron Energy Distribution

The temperature of the electron beam is 10 eV, and the electron velocity distribution follows a Gaussian distribution. Table 2 can be divided into four reaction types, namely: Type-A, Elastic collisions (index: 1 in Table 2); Type-B, Elastic and various excitation collisions (index: 1 and 9–18); Type-C, All collisions (index: 1–19); and Type-D, No collisions. They are applied to the electron transport process in the expansion region containing the magnetic filter field to analyze the electron energy distribution on the extraction surface.

Figure 7 is a diagram of the proportion of electron energy at the extraction surface under the four reaction types when $T = 9 \sim 15$ µs. More relevant parameters of the extracted electrons are shown in Table 3, where the temperature of the extracted electron is derived from the Maxwellian velocity distribution function.

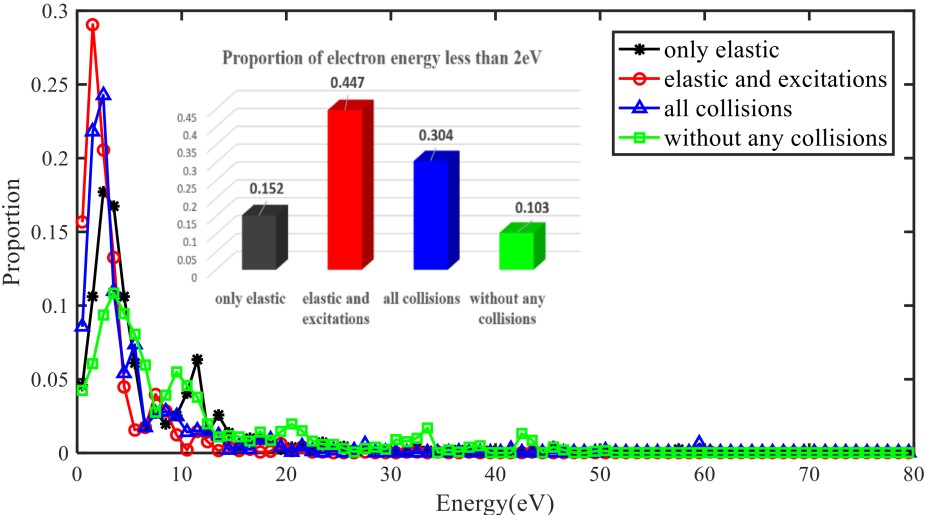

**Figure 7.** Diagram of the proportion of electron energy at the extraction surface under the four types of reaction types when $T = 9 \sim 15$ µs with the emission electron beam at a temperature of 10 eV.

**Table 3.** The related parameters of extracted electrons during $T = 9 \sim 15$ µs.

| Type | Electron Mean Energy $\bar{\varepsilon}$ (eV) | Extracted Electron Temperature $T_{ext}$ (eV) | Extracted Electron Number $N_{ext}$ |
|------|------|------|------|
| A | 6.9 | 4.4 | $1.72 \times 10^{14}$ |
| B | 3.6 | 2.2 | $1.98 \times 10^{14}$ |
| C | 5.3 | 3.2 | $1.02 \times 10^{14}$ |
| D | 10.0 | 6.5 | $5.37 \times 10^{13}$ |

Type-D, i.e., the no collisions model, is considered as the reference model. For Type-A, elastic collisions theoretically cause almost no decay of electron energy. As shown in Table 3, $N_{ext}$ increases by a factor of about three, but $\bar{\varepsilon}$ decreases from 10 eV to 6.9 eV, indicating that elastic collisions promote the mobility of low-energy electrons ($\varepsilon \leq 2$ eV) to a certain extent. For Type-B, $N_{ext}$ is the largest, the proportion of low-energy electrons in Figure 7 is the highest, and $\bar{\varepsilon}$ was the lowest, indicating that excitation collision plays the most important role in the electron energy decay mechanism; for Type-C, $T_{ext} = 3.2$ eV, which is consistent with the experimentally measured 3 eV [31] of a negative hydrogen ion source with the same magnetic field structure. Collision reactions and types in Table 2 basically conform to the main electron collision reactions of the negative hydrogen ion source. The numerical results above show that the magnetic filter field system contributes to the reduction in electron temperature and that electron excitation collision is the main reason for electron energy decay.

Figure 8 shows the spatial distribution of "cold" electrons on the X−Z plane and the average density distribution along the +X direction under the four types of reactions at $T = 15$ μs; the discrete red points represent "cold" electrons.

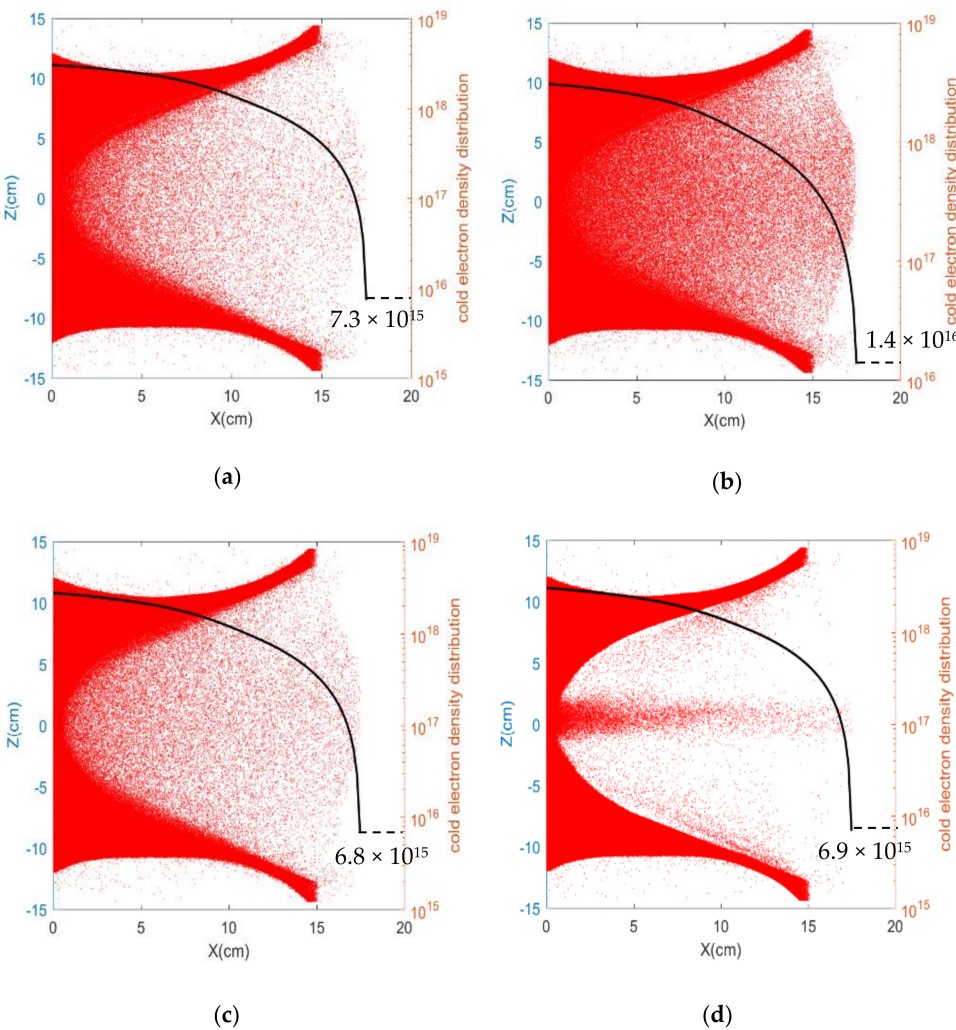

**Figure 8.** The spatial distribution of "cold" electrons on the X−Z plane and the average density distribution along +X direction under four reaction types at $T = 15$ μs. The red dots represent the "cold" electrons ($\varepsilon \leq 2$ eV). (**a**) Type-A. Elastic (index:1); (**b**) Type-B. Elastic and excitations (index:1,9–18); (**c**) Type-C. All collisions (index:1–19); (**d**) Type-D. No collisions.

As shown in Figure 8a,d, elastic collision leads to the change in scattering angle, which makes the diffusion effect of the electrons more pronounced. Results from modelling [29,30] show that a gradient drift occurs in the IPP prototype source. Likewise, the vertical drift of electrons due to gradient drift is observed in Figure 8d. In Figure 8d, there seems to be electron bunching at Z = 0 cm, which is actually the gradient drift of electrons caused by the magnetic field [40]. In all models, magnets are magnetized in the −Z direction, and the direction of $\vec{\nabla B}$ is in the −Y direction. However, since the direction of the magnetic field is parallel to the Z-axis at the central axis Z = 0 cm, the vertical drift of electrons in the −Y direction is more pronounced, creating the phenomenon of "electron bunching" on the X−Z plane as shown in Figure 8d. It can be seen from Figure 8b that the number of "cold" electrons is the largest. In addition, the density of "cold" electrons is also highest at the BP (i.e., X = 17.5 cm, the end of the solid black line), which is largely due to the electron energy decay caused by excitation collisions. Compared with Figure 8b,c, the number of "cold" electrons in the Type-C group is fewer, and the density of "cold" electrons at BP is lower.

It can be seen from Table 3 that $N_{ext}$ in the Type-C group is nearly half that of the Type-B group, and $T_{ext}$ is higher. The reason is that, although the ionization reactions (index: 2, 3, 4 in Table 2) can generate some new electrons, the new electrons share the energy remaining after subtracting the threshold from the original electrons, so the overall energy trend still decreases. However, indexes No. 5, 6, and 7 are the dominant electron destructive reactions, and No. 7 in particular is the volume production of $H^-$ ions, so electrons die out further after energy decay. This indicates that the extracted electron temperature $T_{ext}$ is less affected by ionization but more by excitation and electron destructive reactions.

## 4. Conclusions

Many examples in the existing literature indicate that the magnetic filter field system of RF negative hydrogen ion sources reduces the electron temperature and "filters" or "reflects" the "hot" electrons to avoid the stripping loss of $H^-$ ions [17–20]. However, the numerical results show that the magnetic filter field filters both "hot" and "cold" electrons, which is different from the descriptions in the literature. The numerical results show that the reflection of the electrons is caused by the magnetic mirror effect in the source. The filtering of electrons is reflected in two aspects. For one thing, most of the electrons in the expansion region are still constrained by the magnetic filter field and move along the magnetic field lines to the chamber wall. It is difficult for them to reach the extraction surface. Therefore, the magnetic filter field can reduce the temperature of the extracted electrons to a certain extent. For another thing, collisions lead to the decay of electron energy in the expansion region, where excitation collision is the dominant decay mechanism for electron energy. These two aspects together contribute to lowering the temperature of the electrons at the extraction surface.

In this paper, the mechanism of the magnetic filter field on the "hot" electrons in the expansion region is studied based on the CHIPIC code. The filtering, reflection, and drift phenomena of electrons in the numerical results are presented and discussed to analyze the complex behavior of electrons in a magnetic filter field. The research in this paper helps elucidate the particle transport process under the magnetic filter field system and the objective role of the magnetic filter field system in negative hydrogen ion source devices. The numerical results help to explain the formation process of cold electrons on the extraction surface, thus providing a reference for reducing the loss probability of $H^-$ ions.

**Author Contributions:** Conceptualization, M.X. and H.W.; methodology, D.L.; software, M.X., H.W., D.L. and L.L.; validation, M.X.; formal analysis, M.X., H.W., D.L. and L.L.; investigation, M.X., H.W., D.L. and L.L.; resources, M.X., H.W., D.L. and L.L.; writing—original draft preparation, M.X.; writing—review and editing, M.X. and D.L.; visualization, M.X., H.W., D.L. and L.L.; supervision, M.X., H.W., D.L. and L.L.; project administration, M.X., H.W., D.L. and L.L. All authors have read and agreed to the published version of the manuscript.

**Funding:** This research was funded by NATIONAL NATURAL SCIENCE FOUNDATION OF CHINA, grant number 11905026 and 12075051. This research was funded by NATIONAL KEY RESEARCH AND DEVELOPMENT PROGRAM OF CHINA, grant number 2018YFF01013001.

**Institutional Review Board Statement:** Not applicable.

**Informed Consent Statement:** Not applicable.

**Data Availability Statement:** Data sharing not applicable.

**Conflicts of Interest:** The authors declare no conflict of interest.

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
