# Peer review of "Study on the Correlation between Magnetic Field Structure and Cold Electron Transport in Negative Hydrogen Ion Sources"

_applsci, doi:10.3390/app12094104_

Round 1

Reviewer 1 Report

Your manuscript could be accepted for publication should you be prepared to incorporate moderate revisions.

Reviewer 2 Report

In general terms, the following comments:

  1. The text presents a confusing reading, since the punctuation marks are not correct, it is necessary to rewrite the ideas and improve the language, that is, more formal.
  2. The results presented are correct, but the paper is only written at a descriptive level, it is necessary to go deeper into the physical discussion of these results.
  3. The conclusions do not contribute to the investigation, as already mentioned, they are only descriptive. For example, the first conclusion shown cannot refer to what other investigations establish, this must be justified in the introduction.

More specific examples of wording would be:

In the line 10 and 11 “the mechanism of the magnetic filter system in the expansion region of the negative hydrogen ion source on the "hot" electrons”, must be changed to, “the mechanism of the magnetic filter system, in the expansion region of the negative hydrogen ion source, on the "hot" electrons”.

In the line 17 “The filtering of "hot" electrons is due to the facts that the magnetic field confines most of the electrons to 16 reach the vicinity of the extraction surface and collisions cause the decay of electron energy”, must be changed to, “The filtering of "hot" electrons is due to the facts that the magnetic field confines most of the electrons to reach the vicinity of the extraction surface, being the collisions cause of the decay in electron energy”.

In this way, you have to correct the entire text so that it has more coherence and better strings together the ideas.

On line 46 say “many references show that the negative hydrogen ion source needs to 46 add a filter magnetic field…”, but they only show a reference, you have to broaden the search for references.

On line 56 and 57 say “It is helpful for people to understand the negative hydrogen ion source 56 device and the particle transport process under the filter magnetic field system”, this is a very vague idea, it is necessary to discuss further why it is important.

In some parts of the text they write “hot”, between quotation marks, and in others hot, without quotation marks, the text must be homogenized.

In figure 1 H- is placed, while in the main body H- is placed, with superscript, it is important to have the same nomenclature.

On line 77 say “the yellow represents the emission surface”, when the yellow color is only lines, to which a surface cannot be associated, in their case it should be “the yellow lines represent the ends of the emission surface”.

It is not correct to call the equations as formulas, a formula does not correspond to an equality.

Reviewer 3 Report

1-english needs revision

2-the innovations of the study have to be clarified

3- the technological impact of the present study has to be better explained with respect to the start of art

4-more references are necessary in order to show the real contributions of the present investigation

5-experimental results should be presented in order to corroborate the results  of the calculations

Round 2

Reviewer 2 Report

Comments and observations were satisfactorily addressed by the authors.

Author Response

Thank you for your affiramation of the correctness of the results. We have made minor changes on the manuscript again to broaden the references.